# Entanglement spreading and quasiparticle picture beyond the pair structure

**Alvise Bastianello[1]⋆ and Mario Collura[2]**

**1** Institute for Theoretical Physics, University of Amsterdam,
Science Park 904, 1098 XH Amsterdam, The Netherlands
**2** SISSA, Via Bonomea 265, I-34136 Trieste, Italy

⋆ a.bastianello@uva.nl

## Abstract

The quasi-particle picture is a powerful tool to understand the entanglement spreading in many-body quantum systems after a quench. As an input, the structure of the excitations' pattern of the initial state must be provided, the common choice being pairwise-created excitations. However, several cases exile this simple assumption. In this work we investigate weakly-interacting to free quenches in one dimension. This results in a far richer excitations' pattern where multiplets with a larger number of particles are excited. We generalize the quasi-particle ansatz to such a wide class of initial states, providing a small-coupling expansion of the Rényi entropies. Our results are in perfect agreement with iTEBD numerical simulations.


# 1 Introduction

One of the most fundamental questions in physics is how collective statistical features emerge from a microscopic deterministic time evolution, both in the case where the model at hand is classical or quantum. In particular, recent times witnessed an outburst of interest in the dynamics of closed quantum many-body systems: the rise of new experimental techniques on the one hand [1], as well as the sharpening of our understanding on the theoretical side, laid the foundations of a fruitful synergy among experimentalists and theoreticians. Indeed, the possibility of realizing almost isolated quantum systems and tuning their effective dimensionality, together with a high-precision manipulation of their interactions, gave theoreticians the perfect arena to study their favorite models. Among the various out-of-equilibrium protocols, a prominent role is covered by the *quantum quench* [2]: in its simplicity, it provides a neat and clear framework to ask several fundamental questions. Focusing on homogeneous sudden quenches, one imagines the system to be prepared in a given steady state of an initial Hamiltonian $H_0$: common choices for the initial conditions are the ground state of $H_0$ or, more generally, other physically-motivated states or density matrices (e.g. thermal ensembles). Then, at time $t = 0$ the system is suddenly brought out-of-equilibrium by means of a sudden change of the Hamiltonian $H_0 \rightarrow H$, which is then kept constant: after a long time, despite the non-dissipative (and thus unitary) dynamics, the system is expected to locally relax. The study of the final steady state, and how it is approached, have been at the center of an intensive research (see Refs. [3–6] for a review).

In this respect, low-dimensional systems, nowadays commonly realized in experiments [1, 3, 7–23], are an essential tool for theoreticians in view of the several exact analytical (and numerical [24, 25]) methods, able to capture a pletora of non-perturbative features, as thermalization (or lack of it). Indeed, the one-dimensional world includes integrable [26] and conformal [27] models: despite being interacting and strongly correlated, these models are *exactly solvable* and, because of this, they have been at the center of an intensive research, providing an impressive array of results [5].

Among the various achievements, the late time relaxation of these models has been shown to be described by a Generalized Gibbs Ensemble [28] rather than the common thermal one (see Ref. [5] and reference therein). Very recently, weakly inhomogeneous and time-dependent protocols have been made accessible through the Generalized Hydrodynamics [29–32]. Furthermore, spreading of both correlations and entanglement has been framed in a simple, yet powerful, quasi-particle picture [33]. In particular, how to generalise this ansatz is at the focus of our investigation.

To be concrete, let us consider a many-body quantum system, e.g. a spin chain, initialized in a certain pure state $|\psi\rangle$. After dividing the system in two parts $A \cup B$, we may wonder how much the two are entangled: this can be measured by the von Neumann entanglement entropy [34–36]

$$S_A(t) = -\text{Tr}_A \hat{\rho}_A(t) \log \hat{\rho}_A(t), \tag{1}$$

where we defined the reduced density matrix of the subsystem $A$, $\hat{\rho}_A(t) = \text{Tr}_B |\psi(t)\rangle \langle \psi(t)|$, obtained by tracing out the degrees of freedom within the $B$ part of the time-evolved state $|\psi(t)\rangle$. Together with the von Neumann entropy, other common quantifiers of entanglement are the Rényi entropies, defined for arbitrary real indexes $N$ as

$$S_A^{(N)}(t) = \frac{1}{1-N} \log \text{Tr}_A \hat{\rho}_A^N(t). \tag{2}$$

Above, $N$ can be any real number, but later on we restrict to the case of integers $N$. In general, entanglement entropies are not easy to compute: apart from free systems [37], progresses can be made in critical systems taking advantage of conformal invariance [38, 39] with extensions

to out-of-equilibrium setups [33] and recently to inhomogeneous cases [40–44]. In particular, bringing a low-entanglement state out-of-equilibrium will result in the entanglement growing with time: a powerful method to describe (the scaling part of) the entanglement spreading is provided by the *quasi-particle picture*.

The scaling part of the entanglement growth is defined as it follows. Let the bipartition $A \cup B$ be identified by a set of coordinates $x_0 < x_1 < ...$ in such a way $A = \cup_{i=0}[x_{2i}, x_{2i+1}]$ and $B = \cup_{i=1}[x_{2i+1}, x_{2i+2}]$, then we define a new bipartition $A^{(\lambda)} \cup B^{(\lambda)}$ with rescaled coordinates $A^{(\lambda)} = \cup_{i=0}[\lambda x_{2i}, \lambda x_{2i+1}]$ ($B^{(\lambda)}$ is analogously defined). The scaling part of the entanglement is defined as

$$\mathcal{S}_A^{(N)}(t) = \lim_{\lambda \to \infty} \lambda^{-1} S_{A^{(\lambda)}}^{(N)}(\lambda t). \tag{3}$$

Despite not being an ab-initio calculation, the quasi-particle picture is able to exactly reproduce several features of the entanglement growth after a quantum quench. Originally introduced in critical systems [33, 45, 46], its applicability has been promptly extended first to generic non-interacting models [47–54], then to interacting integrable ones [55–60] (with recent developments in non-homogeneous systems [61–63]); finally, it has provided important insight even into not strictly-integrable systems [64, 65].

The idea is based on the following semiclassical argument: let us imagine to perform a homogeneous quantum quench. At time $t = 0^+$ the system is brought out-of-equilibrium and, from the perspective of the post-quench Hamiltonian, the system lays in a highly excited state. Semiclassically, one can depict such a state as a gas of excitations, which then start spreading across the system. If the post-quench model is a CFT or an integrable model, these excitations are stable and undergo a ballistic motion. If instead the state is evolved with a non-integrable dynamics, the quasi-particle are not stable anymore and acquire a finite life-time. For the sake of simplicity, we consider only the case of stable excitations. The traveling particles are ultimately responsible of the entanglement (and correlation) spreading. In this regard, the understanding of the excitations' pattern is of utmost importance in order to drag quantitative predictions.

In the original formulation [33], quasi-particles are assumed to be produced in pairs of opposite momenta: this is the simplest assumption which respects the translational invariance of the protocol, and it is the common situation in most of free-to-free quenches and in exactly solvable quenches in integrable models [66]. This is, for example, the case for the protocol studied in Refs. [67, 68].

For the sake of clarity (and later comparison with previous results), we briefly review free quenches in the Ising spin chain, whose Hamiltonian is given by

$$\hat{H}_I = -\frac{1}{2} \sum_j \left( \sigma_j^x \sigma_{j+1}^x + h \sigma_j^z \right), \tag{4}$$

where $\sigma^{x,y,z}$ are the standard Pauli matrices and the system is assumed infinitely large. By mean of a Jordan-Wigner transformation, the Hamiltonian (4) is readily mapped into a free-fermion model which is then diagonalized in the Fourier space with modes $\{\eta(k), \eta^\dagger(q)\} = \delta(k-q)$

$$\hat{H}_I = \int_{-\pi}^{\pi} dk \, \omega(k) \eta^\dagger(k)\eta(k) + \text{const.} \tag{5}$$

We will thoroughly analyze the solution of the Ising model in Section 2. Initializing the system in the ground state of the Hamiltonian for a certain magnetic field $\bar{h}$ and quenching $\bar{h} \to h$, one can express the pre-quench state in terms of the post-quench modes in the form of a squeezed state

$$|\psi\rangle = \frac{1}{\sqrt{\mathcal{N}}} \exp\left[\frac{1}{2} \int_{-\pi}^{\pi} dk \, \mathcal{K}_2(k,-k)\eta^\dagger(k)\eta^\dagger(-k)\right]|0\rangle, \tag{6}$$

where $\mathcal{N}$ is a normalization factor, and $\mathcal{K}_2(k,-k)$ is a non-trivial function of $\bar{h}$ and $h$. The state $|0\rangle$ is the ground state (or vacuum) of the post quench Hamiltonian $\hat{H}_I$, i.e. $\eta(k)|0\rangle = 0$. As the form in Eq. (6) suggests, excitations are indeed independently created in pairs, their density being fixed by the mode density $\langle \eta^\dagger(k)\eta(q)\rangle = \delta(k-q)n(k)$

$$n(k) = \frac{|\mathcal{K}_2(k,-k)|^2}{1+|\mathcal{K}_2(k,-k)|^2}. \tag{7}$$

Within the quasi-particle picture, backed by the assumption of the pair structure, the initial state is regarded as a source of paired excitations: pairs created at different spatial points or belonging to different momenta are uncorrelated and disentangled. Only excitations belonging to the same pair are responsible for the entanglement spreading, the total effect being additive on the different pairs. More precisely, at time $t > 0$ particles ballistically travel with velocity $v(k) = \partial_k \omega(k)$ and only pairs such that one excitation lays in $A$ and the other in $B$ contribute to the entanglement. The Rényi entropies are thus

$$\mathcal{S}_A^{(N)}(t) = \int dx \int_{-\pi}^{\pi} \frac{dk}{2\pi} \chi_A(x + tv(k))\chi_B(x + tv(-k))s^{(N)}(k), \tag{8}$$

where $\chi_A(x)$ is the characteristic function of the subsystem $A$, namely $\chi_A(x) = 1$ if $x \in A$, otherwise zero. The factor $s^{(N)}(k)$ is[1]

$$s^{(N)}(k) = \frac{1}{1-N} \log\left\{[1-n(k)]^N + n(k)^N\right\}. \tag{9}$$

As powerful as the quasi-particle picture is, the pair-structure is a very restrictive constraint: even though free-to-free quenches generally display this pattern, this is not true in more general setups. For example, preparing the system in the ground state of an interacting Hamiltonian (where the Wick theorem does not hold) and quenching to the free dynamics, the state cannot be in the form given by Eq. (6), which does satisfy the Wick theorem.

Hence, one is naturally led to wonder whether the quasi-particle approach can be generalized, dropping the pair-assumption in favor of a more general structure: a first step in this direction has been pursued in Refs. [69, 70].

In particular, Ref. [69] considers quenches in a free model starting from a suitably-engineered initial state, where excitations were created in multiplets which shared classical correlation among the components. As such, the quasi-particle ansatz could be generalized in terms of purely-classical concepts. On the other hand, Ref. [70] considered quenches in free lattice models whose couplings are periodically modulated in space with period $T$, leading to a quasi-particle picture with excitations generated in pairs of momenta $(k_1, k_2)$, with $k_1 + k_2 = 0 \mod 2\pi/T$, but with non-trivial quantum correlation among the different pairs. In both cases, the Wick theorem held on the initial states.

In this paper, we consider the very generic situation where the system is initialized in the ground state of a weakly-interacting Hamiltonian and then quenched to the free point: this generates a complicated pattern of excitations which exile the pair structure, as well as the studies of Refs. [69, 70]. In particular, we consider initial states in the following "generalized" squeezed form [71],

$$|\psi\rangle = \frac{1}{\sqrt{\mathcal{N}}} \exp\left[\sum_{n=1}^{\infty} \frac{1}{(2n)!} \int_{-\pi}^{\pi} d^{2n}k \, \delta\left(\sum_{j=1}^{2n} k_j\right) \mathcal{K}_{2n}(k_1,...,k_{2n}) \prod_{j=1}^{2n} \eta^\dagger(k_j)\right]|0\rangle, \tag{10}$$

---

[1]The von Neumann entanglement entropy is obtained taking the limit $N \to 1$.

where multiplets (or clusters) of more than two particles are produced and the Wick theorem does not apply anymore. Above, we are assuming an underlying lattice, as such the wave-vectors $k$ are confined to a Brillouin zone $[-\pi, \pi]$, but the extension to continuum models is straightforward.

We provide a generalization of the quasi-particle picture for the Rényi entropies of integer index $N$ to these states with a multiplet structure in form of a systematic expansion valid for small quenches (i.e. $\mathcal{K}_{2n}$ are supposed to be small), which perfectly reproduces exact numerical simulations.

The paper is organized as follows: in Section 2 we provide a specific quench protocol which has Eq. (10) as an initial state. The quench consists in turning off a small local interaction in a spin chain, letting then the system to evolve with the Ising Hamiltonian (4). We show how to perturbatively compute the amplitudes $\mathcal{K}_{2n}$ in terms of the quench's parameter. In Section 3 we discuss our generalized quasi-particle picture and test it with iTEBD simulations, finding excellent agreement; whilst, in contrast, the naive application of the method assuming a pair structure gives incorrect results. Section 4 gathers our conclusions, and some technical details are left in the Appendices.

## 2 The model

Albeit our generalized quasi-particle picture can be applied to any initial state in the form Eq. (10), for the sake of clarity it is useful to refer to a simple model which displays such a structure. Hence, we consider an infinitely long spin chain with the following Hamiltonian

$$\hat{H} = -\frac{1}{2}\sum_j \left\{ \sigma_j^x \sigma_{j+1}^x + h\sigma_j^z \right\} - \gamma \sum_j \left\{ \frac{C_1}{4}\sigma_j^z \sigma_{j+1}^z + C_2\sigma_j^z + C_3\sigma_{j+1}^x \sigma_j^x + C_4\sigma_{j+1}^y \sigma_j^y \right\}, \quad (11)$$

where $C_1, C_2, C_3, C_4$ are tunable parameters and $\gamma$ controls the interaction strength: for the time being, we consider arbitrary values for $C_i$. Later on, we tune these constants in order to enhance the role of the multiparticle clusters in Eq. (10) when compared with the pairs. We assume the system to be initialized in the ground state for $\gamma \neq 0$, then at $t = 0$ we switch the interaction off $\gamma = 0$ and let the system evolve with the Ising Hamiltonian. For $C_1 = C_4 = 0$, the protocol reduces to a transverse-field quench in the Ising chain: this free-to-free quench has been already studied in Ref. [68]. Hence, the pair structure holds in this case. On the contrary, if $C_1$ and $C_4$ are not zero, the pre-quench Hamiltonian is truly interacting: the ground state does not satisfy the Wick theorem any longer, and thus the pair structure for the post quench excitations is inevitably spoiled.

Before of discussing the structure of the interacting ground state, we briefly review the diagonalization of the Ising Hamiltonian (4) which governs the post-quench dynamics. First, the spin variables can be written in terms of standard fermionic operators through the following Jordan-Wigner (JW) transformation

$$\sigma_j^+ = \prod_{\ell < j}(1 - 2c_\ell^\dagger c_\ell)c_j, \quad \sigma_j^- = \prod_{\ell < j}(1 - 2c_\ell^\dagger c_\ell)c_j^\dagger, \quad \sigma_j^z = 1 - 2c_j^\dagger c_j, \quad (12)$$

with $\sigma^\pm = (\sigma_x \pm i\sigma_y)/2$. The fermions satisfy standard anticommutation rules $\{c_j, c_{j'}^\dagger\} = \delta_{j,j'}$. After the JW transformation, the Ising Hamiltonian is rewritten in the following form (we drop additive constants which do not affect the dynamics)

$$\hat{H}_I = \sum_j -\frac{1}{2}\left( c_j^\dagger c_{j+1}^\dagger + c_j^\dagger c_{j+1} + c_{j+1}c_j + c_{j+1}^\dagger c_j \right) + hc_j^\dagger c_j. \quad (13)$$

In terms of the fermions, the Ising Hamiltonian is quadratic and promptly diagonalized in the Fourier space, once the proper fermionic modes $\eta(k)$ are defined

$$c_j = \int_{-\pi}^{\pi} \frac{dk}{\sqrt{2\pi}} e^{ikj} \left[ \cos(\Omega_k)\eta(k) + i\sin(\Omega_k)\eta^{\dagger}(-k) \right], \tag{14}$$

where the Bogoliubov angle $\Omega_k$ is

$$\tan\Omega_k = \frac{\omega(k) + \cos k - h}{\sin k}, \tag{15}$$

and $\omega(k) = \sqrt{(\cos k - h)^2 + \sin^2 k}$ are the single-particle energies of the modes entering in the diagonal Ising Hamiltonian (5). Since $\omega(k) > 0$, the ground state can be identified with the vacuum of the theory $|0\rangle$, namely the state annihilated by all the modes $\eta(k)|0\rangle = 0$.

We now move to discuss the pre-quench Hamiltonian (11). We use the JW transformation to recast it in terms of the fermionic degrees of freedom, and then rewrite it in terms of the diagonal mode-operators of the non-interacting Ising Hamiltonian. This tedious, albeit simple, calculation results in the following general form

$$\begin{aligned}
\hat{H} = & \int dk\, \omega(k)\eta^{\dagger}(k)\eta(k) \\
& + \gamma \sum_{n_1,n_2} \int_{-\pi}^{\pi} \frac{d^{n_1}k\, d^{n_2}q}{n_1! n_2!} \delta\left( \sum_{i=1}^{n_1} k_i - \sum_{j=1}^{n_2} q_j \right) \mathcal{H}_{n_1,n_2}(k_1,...,k_{n_1}|q_1,...,q_{n_2}) \\
& \times \prod_{i=1}^{n_1} \eta^{\dagger}(k_i) \prod_{j=1}^{n_2} \eta(q_j),
\end{aligned} \tag{16}$$

where, the summation is over all the positive integers $n_1$ and $n_2$, and the coefficients $\mathcal{H}_{n_1,n_2}$ are complicated functions of the pre-quench parameters $C_i$ and of the magnetic field $h$. The $\delta-$function in the momentum space is a consequence of the translational symmetry of the problem. The coefficients $\mathcal{H}_{n_1,n_2}$ must satisfy certain constraints discussed hereafter. From the anti-commutation relations of the fermions we have that $\mathcal{H}_{n_1,n_2}(k_1,...,k_{n_1}|q_1,...,q_{n_2})$ is an antisymmetric function of the $k$ and $q$ momenta separately. The hermicity of the Hamiltonian (16) implies

$$\mathcal{H}_{n_1,n_2}(k_1,...,k_{n_1}|q_1,...,q_{n_2}) = \mathcal{H}^*_{n_2,n_1}(q_{n_2},...,q_1|k_{n_1},...,k_1). \tag{17}$$

Moreover, for parity reasons, the coefficient $\mathcal{H}_{n_1,n_2}$ vanishes whenever $n_1 + n_2$ is odd. The task of computing the coefficients $\mathcal{H}_{n_1,n_2}$ in terms of the interactions in Eq. (11) is straightforward. For example, in the simplest case of a quench among two Ising spin chains, namely posing $C_1 = C_3 = C_4 = 0$ and $C_2 = 1$, one obtains

$$\mathcal{H}_{1,1}(k|k) = \cos(2\Omega_k), \quad \mathcal{H}_{2,0}(k,-k|) = -i\sin(2\Omega_k), \tag{18}$$

and all the other coefficients are zero. In the more general case with $C_1, C_3$ and $C_4$ not zero, also the coefficients $\mathcal{H}_{n_1,n_2}$ with $n_1 + n_2 = 4$ are present, while all the terms with $n_1 + n_2 > 4$ vanish.

We will explicitly write the expression for $\mathcal{H}_{n_1,n_2}$ for a specific choice of the parameters $C_i$ after having discussed the structure of the ground state in the forthcoming section.

## 2.1 Multiplets structure of the pre-quench state

There are several reasons that point to the structure in Eq. (10) for the pre-quench state in terms of the post-quench excitations, all of them extensively discussed in Ref. [71]. Such a state

satisfies, at any order in a small $\mathcal{K}$ expansion, basic thermodynamic properties that ground states of a local Hamiltonian must have; namely, the extensive behavior of thermodynamic quantities and the cluster property for local observables [71]. In addition, the form in Eq. (10) can be recovered from standard perturbation theory. We thus assume the ground state to be in this form and discuss how to determine the coefficients $\mathcal{K}_n$ starting from the Hamiltonian in Eq. (16): we follow the same method presented in Ref. [71], namely we impose the eigenvalue equation $\hat{H}|\psi\rangle = E_G|\psi\rangle$, together with the ansatz in Eq. (10). In this respect, one should notice that applying an annihilation operator $\eta(p)$ to the state (10) is equivalent to take a functional derivative of the exponential, namely

$$\eta(p)|\psi\rangle = \frac{1}{\sqrt{\mathcal{N}}}\frac{\delta}{\delta\eta^\dagger(p)}\exp\left[\sum_{n=1}^{\infty}\frac{1}{(2n)!}\int_{-\pi}^{\pi}\mathrm{d}^{2n}k\,\delta\left(\sum_{j=1}^{2n}k_j\right)\mathcal{K}_{2n}(k_1,...,k_{2n})\prod_{j=1}^{2n}\eta^\dagger(k_j)\right]|0\rangle$$

$$= \left[\sum_{n=1}^{\infty}\frac{1}{(2n-1)!}\int_{-\pi}^{\pi}\mathrm{d}^{2n-1}k\,\delta\left(p+\sum_{j=1}^{2n-1}k_j\right)\mathcal{K}_{2n}(p,k_1,...,k_{2n-1})\prod_{j=1}^{2n-1}\eta^\dagger(k_j)\right]|\psi\rangle. \quad (19)$$

Above, while taking the functional derivatives, the anticommutation relations of the fermions impose that the field $\eta^\dagger$ must be regarded as a Grassmanian variable, leading to extra signs. For example, one has

$$\frac{\delta}{\delta\eta^\dagger(p)}[\eta^\dagger(k_1)\eta^\dagger(k_2)] = \delta(p-k_1)\eta^\dagger(k_2) - \delta(p-k_2)\eta^\dagger(k_1). \quad (20)$$

Therefore, replacing the annihilation operators in Eq. (16) with the functional derivatives, and applying the Hamiltonian on $|\psi\rangle$, one reaches the following set of equations

$$\left\{\sum_{n=1}^{\infty}\int_{-\pi}^{\pi}\mathrm{d}^{2n}k\,\delta\left(\sum_{j=1}^{2n}k_j\right)\left[\left(\sum_{j=1}^{2n}\omega(k_j)\right)\frac{1}{(2n)!}\mathcal{K}_{2n}(k_1,...,k_{2n})\right.\right.$$

$$\left.\left.+\frac{\gamma}{(2n)!}\mathcal{H}_{2n,0}(k_1,...,k_{2n}|)+\gamma\mathcal{I}_{2n}[\mathcal{K}](k_1,...,k_{2n})\right]\prod_{j=1}^{2n}\eta^\dagger(k_j)\right\}|\psi\rangle = E_G|\psi\rangle, \quad (21)$$

where $\mathcal{I}_{2n}[\mathcal{K}]$ are non-trivial functions of $\mathcal{K}_{2n}$ which couple multiplets with different indexes and vanish if one sets $\mathcal{K} = 0$. The computation of $\mathcal{I}_{2n}$ can be framed into a Feynman diagrams picture [71] which, due to its technical aspect, we leave to Appendix A. The condition (21) can be satisfied if and only if the infinite set of equations

$$\left(\sum_{j=1}^{2n}\omega(k_j)\right)\frac{1}{(2n)!}\mathcal{K}_{2n}(k_1,...,k_{2n})+\frac{1}{(2n)!}\gamma\mathcal{H}_{2n,0}(k_1,...,k_{2n}|)+\gamma\mathcal{I}_{2n}[\mathcal{K}](k_1,...,k_{2n}) = \delta_{n,0}E_G$$

$$(22)$$

is imposed. In general, even in the case where a finite number of $\mathcal{H}_{n_1,n_2}$ appears in the Hamiltonian (16), these equations cannot be closed with a finite number of $\mathcal{K}_{2n}$ wavefunctions. The free-to-free case is exceptional and the equations can be closed with only $\mathcal{K}_2$ being not zero (as it should be): in Appendix A we analyze this case and obtain the exact $\mathcal{K}_2$ amplitude, which can be computed by mean of other methods, thus providing a consistency check.

Here, we rather approach the problem in the small interaction limit: within this assumption, Eq. (22) is extremely useful in developing a $\gamma$−power expansion for the $\mathcal{K}$−amplitudes by means of an iterative solution.

This perturbative expansion differs from the standard ground-state perturbation theory [72]. Even retaining only up to a finite order in the $\gamma$−expansion of the functions $\mathcal{K}_{2n}$, contributes to any order in $\gamma$ in the expansion of the state $|\psi\rangle$ are present, because of the series

expansion of the exponential in Eq. (10). In other words, one can look at the $\gamma-$expansion of the functions $\mathcal{K}_{2n}$ as a partial re-summation of the standard perturbation theory on the non-interacting ground state. As already mentioned, among the advantages of the present approach there is the fact that at any perturbative order the state retains some features that a proper ground state should have [71]; namely the extensive behavior of thermodynamics quantities, and the cluster property of the correlation functions. From Eq. (22), one gets

$$\mathcal{K}_{2n}(k_1,...,k_{2n}) = -\gamma \frac{\mathcal{H}_{2n,0}(k_1,...,k_{2n}|)}{\sum_{j=1}^{2n} \omega(k_j)} + \mathcal{O}(\gamma^2). \tag{23}$$

Hence, the coefficients $\mathcal{H}_{2n,0}$ determine, at first order in $\gamma$, the multiplets' population. As we previously commented, setting $C_1 = C_3 = C_4 = 0$ in the Hamiltonian (11) results in a squeezed state (6): consistently, $\mathcal{H}_{2n\neq2} = 0$ and Eq. (23) predicts a non-zero amplitude $\mathcal{K}_2$, while all other terms vanish. On the other hand, the coefficients $C_i$ can be suitably engineered to suppress the pair production and enhance other multiplets.

In particular, feeding the mode decomposition (14) into the general Hamiltonian (11) (after it has been expressed in the fermionic basis through the Jordan-Wigner transformation) and rearraging the whole expression in the form (16), one can impose $\mathcal{H}_{2,0} = 0$ choosing

$$C_1 = 1, \qquad C_2 = -\frac{1 - 2\langle c_j^\dagger c_j \rangle_0}{2}, \tag{24}$$

$$C_3 = \frac{\langle c_{j+1}^\dagger c_j^\dagger \rangle_0 - \langle c_j c_{j+1}^\dagger \rangle_0}{2}, \qquad C_4 = -\frac{\langle c_j c_{j+1}^\dagger \rangle_0 + \langle c_{j+1}^\dagger c_j^\dagger \rangle_0}{2}. \tag{25}$$

Above, $\langle...\rangle_0$ is the expectation values of the fermionic operators on the non-interacting ground state, namely

$$\langle c_j^\dagger c_j \rangle_0 = \int_{-\pi}^{\pi} \frac{\mathrm{d}k}{2\pi} \sin^2 \Omega_k, \tag{26}$$

$$\langle c_{j+1}^\dagger c_j^\dagger \rangle_0 = -\int_{-\pi}^{\pi} \frac{\mathrm{d}k}{2\pi} \sin k \sin \Omega_k \cos \Omega_k, \tag{27}$$

$$\langle c_j c_{j+1}^\dagger \rangle_0 = \int_{-\pi}^{\pi} \frac{\mathrm{d}k}{2\pi} \cos k \cos^2 \Omega_k. \tag{28}$$

In this case, all the $\mathcal{H}_{2n\neq4,0}$ coefficients vanish, while

$$\mathcal{H}_{4,0}(k_1,k_2,k_3,k_4) = \sum_P \mathrm{sign}(P) \frac{e^{ik_{P(4)}+ik_{P(3)}}}{2\pi} \cos \Omega_{k_{P(1)}} \sin \Omega_{k_{P(2)}} \cos \Omega_{k_{P(3)}} \sin \Omega_{k_{P(4)}}. \tag{29}$$

Therefore, this quench creates only multiplets of 4 particles (at first order in $\gamma$). Above, the sum is over all the possible permutations $P$ of four elements.

## 3  The generalized quasi-particle picture for the entanglement growth

After having discussed the structure of the pre-quench state in terms of the post-quench excitations, we can now generalize the quasi-particle picture to include the presence of multiplets beyond the simpler pairs' case. It should be stressed that the quasi-particle picture is an ansatz: its formulation is built on reasonable assumptions and on the experience gained from the previous literature, which also provides an important consistency check. However, its correctness

must be ultimately confirmed by numerical simulations, which turn out to be in perfect agreement with the proposed ansatz.

Similarly to the case with paired excitations, we look at the initial state as a homogeneous source of quasi-particles (but generalizations to inhomogeneous cases are straightforward [62]), which organize in multiplets as Eq. (10) suggests. In this respect, following Refs. [62,70], we introduce a set of auxiliary Hilbert spaces $\mathcal{H}_x$, where $x$ labels the position (in the perspective of a coarse grain analysis, we can replace the underlying lattice with a continuum). Similarly, we introduce auxiliary fermionic mode operators $\{\eta_x(k), \eta_x^\dagger(q)\} = \delta(k-q)$ and a local vacuum $|0_x\rangle$, then we promote Eq. (10) to a state $|\psi_x\rangle$ in the local Hilbert space $\mathcal{H}_x$

$$|\psi_x\rangle = \frac{1}{\sqrt{\mathcal{N}}} \exp\left[\sum_{n=1}^\infty \frac{1}{(2n)!} \int_{-\pi}^\pi d^{2n}k\, \delta\left(\sum_{j=1}^{2n} k_j\right) \mathcal{K}_{2n}(k_1,...,k_{2n}) \prod_{j=1}^{2n} \eta_x^\dagger(k_j)\right] |0_x\rangle. \quad (30)$$

Above, the amplitudes $\mathcal{K}_{2n}$ are those of the homogeneous model. The quasi-particle ansatz states that excitations created at different points in space are uncorrelated and disentangled, as such they contribute additively to the entanglement: using of the auxiliary Hilbert spaces, we write

$$\mathcal{S}_A^{(N)}(t) = \int dx\, s_{\mathcal{A}_{x,t}}^{(N)}, \quad (31)$$

where $s_{\mathcal{A}_{x,t}}^{(N)}$ is the entanglement density contribution coming from each $|\psi_x\rangle$ state. We are considering bipartitions of the system $A \cup B$ in the real space, but the local Hilbert spaces are constructed in terms of creation-annihilation operators with the momentum $k$ as quantum number. The correct dictionary to translate the bipartion $A \cup B$ into a proper bipartition in the momentum space of the local auxiliary Hilbert space relies on the ballistic propagation of the quasi-particles

$$k \in \mathcal{A}_{x,t} \quad \Longleftrightarrow \quad x + t v(k) \in A, \quad (32)$$

and $\mathcal{B}_{x,t}$ is analogously defined. In Fig. 1 we provide a semiclassical representation of Eq. (32). The contribution to the entanglement entropy of each spatial point is defined tracing on the local degrees of freedom. In this respect, we define the reduced density matrix

$$\hat{\rho}_{\mathcal{A}_{x,t}} = \text{Tr}_{\mathcal{B}_{x,t}} |\psi_x\rangle\langle\psi_x| \quad (33)$$

and then the entanglement entropy. While doing so, a subtlety arises: indeed, the entanglement entropy coming from Eq. (33) is formally divergent (as we discuss in Appendix B) and its computation requires a proper finite-volume regularization

$$\frac{\ell}{2\pi} \int dk \to \sum_k, \quad \text{with} \quad \frac{\ell}{2\pi} k \in \mathbb{Z}, \quad (34)$$

with $\ell$ playing the role of a large (but finite) volume. Then, one defines the entanglement density $s_{\mathcal{A}_{x,t}}^{(N)}$ as

$$s_{\mathcal{A}_{x,t}}^{(N)} = \frac{1}{1-N} \lim_{\ell \to \infty} \left[\ell^{-1} \log \text{Tr}_{\mathcal{A}_{x,t}} \hat{\rho}_{\mathcal{A}_{x,t}}^N\right]. \quad (35)$$

In order for the quasi-particle picture to be predictive, one is left with the highly challenging task of computing $s_{\mathcal{A}_{x,t}}^{(N)}$. In general, the structure of Eq. (30) when other multiplets besides pairs are present is too much complicated to drag an exact result. However, in the limit of small quenches (i.e. $\mathcal{K}_{2n}$ small), one can revert to a perturbative approach and expand the Rényi entropies for integer indexes $N$ in powers of the amplitudes $\mathcal{K}_{2n}$. We present the full

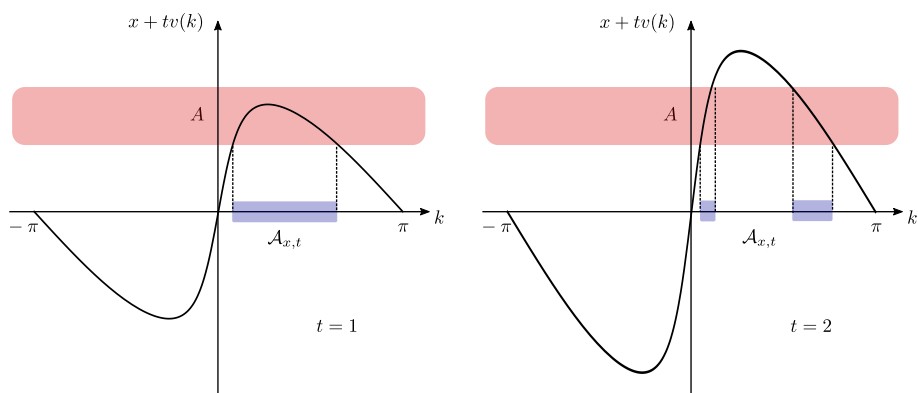

Figure 1: Pictorial representation of the bipartition induced in the auxiliary Hilbert space $\mathcal{H}_x$. On the horizontal axis we consider the momenta within the first Brillouin zone, on the vertical axis we consider the function $x + tv(k)$. We focus on two different times. We consider a bipartition in the real space $A \cup B$ with $A$ an interval in the vertical axis, which is then prolongated in the whole red-shaded area. The domain $\mathcal{A}_{x,t}$ (blue-shaded area) in the momentum space is determined by those momenta such that $x + tv(k)$ lays within the interval $A$.

technique, based on Feynman diagrams, in Appendix B: hereafter, we provide and discuss the first non-trivial order and compare with ab-initio numerical simulations of the model (11). For $s^{(N)}_{\mathcal{A}_{x,t}}$ with $N \in \mathbb{N}$, one finds

$$
s^{(N)}_{\mathcal{A}_{x,t}} = \frac{N}{N-1} \frac{1}{2\pi} \sum_n \sum_{j=1}^{2n-1} \frac{1}{j!(2n-j)!} \int_{\mathcal{A}_{x,t}} \mathrm{d}^j k \int_{\mathcal{B}_{x,t}} \mathrm{d}^{2n-j} q \, |\mathcal{K}_{2n}(k_1, ..., k_j, q_1, ..., q_{2n-j})|^2
$$
$$
\times \delta \left( \sum_{i=1}^{j} k_i + \sum_{i=1}^{2n-j} q_i \right) + \mathcal{O}(\mathcal{K}^3). \tag{36}
$$

As a simple consistency check, one can consider the case where only pairs are present and compare with the usual quasi-particle ansatz (8), using the mode-density (7) for small $\mathcal{K}_2$: of course, the two calculations are in complete agreement. As we have already stressed, Eq. (36) holds for integers $N$, but one could wonder if the result can be analytically continued to real indexes and, through the limit $N \to 1$, reaching the von Neumann entanglement entropy. Unfortunately, this is not the case: dropping further corrections in $\mathcal{K}$ and promoting $N$ to be a real variable, the limit $N \to 1$ of Eq. (36) diverges. On the other hand, $\lim_{N \to 1} s^{(N)}_{\mathcal{A}_{x,t}}$ is expected to exists and to give the quasi-particle picture for the von Neumann entropy. This discrepancy is due to the fact that the operation of truncating the $\mathcal{K}$−expansion does not commute with the analytic continuation: there could be terms, which we are neglecting in the Rényi entropies, such that after a proper analytic continuation become of the same order of, or even more relevant than, the terms written explicitly in Eq. (36).

Now, we specialize the discussion to the model of Section 2 and compare the quasiparticle prediction with iTEBD simulations. For general values of the couplings $C_i$ in Eq. (11), both pairs and quartuplets are generated at the first order in the interaction (23): higher clusters of particles are created as well, but at further orders in the perturbation theory. Hence, Eq. (36) predicts an entanglement growth $\propto \gamma^2$ (with further corrections $\mathcal{O}(\gamma^3)$) where pairs and quartuplets contribute additively. In order to enhance the role of the quartuplets, we now consider the choice of parameters Eq. (24) which, at the first order in $\gamma$, does not excite pairs.

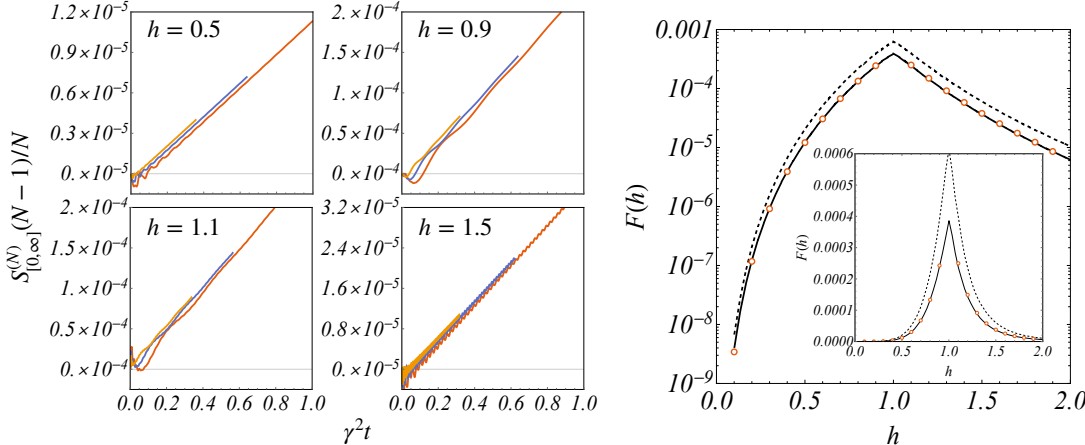

Figure 2: (**Left panels**) iTEBD evolution of the Rényi entropies for $N = 4$ and four representative values of the transverse field $h$. All curves have been vertically shifted by subtracting the corresponding ground-state entropy at $t = 0$. Different colors represent different quenches from $\gamma = 0.1$ (red), $0.08$ (blue), $0.06$ (yellow) to the noninteracting theory $\gamma = 0$. (**Right panel**) The numerical slope of the Rényi entropies in log scale (inset in linear scale), obtained through a linear fit of the time-dependent data sets (symbols), perfectly matches the analytical multiplet prediction (full black line). The black dashed line is the incorrect prediction based on the pair-structure ansatz of the initial state. The details on the iTEBD simulations can be found in App. C.

We first focus on a bipartition of the system where $A = [0, \infty]$: in this case, the integrals and summations appearing in Eq. (36) can be greatly simplified. Indeed, considering a multiplet of quasiparticles $\{k_i\}_{i=1}^4$ which ballistically propagates for a time $t$, it can contribute to the entanglement if and only if the excitations were created at position $x \in (t \min_{k_i} v(k_i), t \max_{k_i} v(k_i))$. This causes the entanglement to linearly grow with time. More precisely, one gets the following scaling form

$$\mathcal{S}_{[0,\infty]}^{(N)} = \frac{N}{N-1} t\left[\gamma^2 F(h) + \mathcal{O}(\gamma^3)\right], \tag{37}$$

with

$$F(h) = \frac{1}{2\pi} \frac{1}{4!} \int_{-\pi}^{\pi} \mathrm{d}^4 k \, \delta\left(\sum_{i=1}^4 k_i\right)\left[\max_{k_i} v(k_i) - \min_{k_i} v(k_i)\right]\left|\frac{\mathcal{H}_{4,0}(k_1, k_2, k_3, k_4|)}{\sum_{i=1}^4 \omega(k_i)}\right|^2. \tag{38}$$

In Fig. 2 we provide a numerical check of the scaling form for different Rényi entropies, together with a comparison of the slope $F(h)$ computed as per Eq. (38) with the same quantity extracted from iTEBD results. In order to stress the importance of considering the multiplets, we plot the slope $F(h)$ one would find expanding Eq. (9) and Eq. (8) at the same order in $\gamma^2$, using for $n(k)$ the mode density computed on the prequench state:

$$n(k) = \frac{\gamma^2}{3!} \int_{-\pi}^{\pi} \mathrm{d}^3 q \left|\frac{\mathcal{H}_{4,0}(k, q_1, q_2, q_3|)}{\omega(k) + \sum_{j=1}^3 \omega(q_j)}\right|^2 \delta(k + q_1 + q_2 + q_3) + \mathcal{O}(\gamma^3). \tag{39}$$

The two curves are clearly apart and the numerical data are in perfect agreement with the multiplet result. In Fig. 3, instead, we consider a finite interval $A = [0, L]$: in this case, the

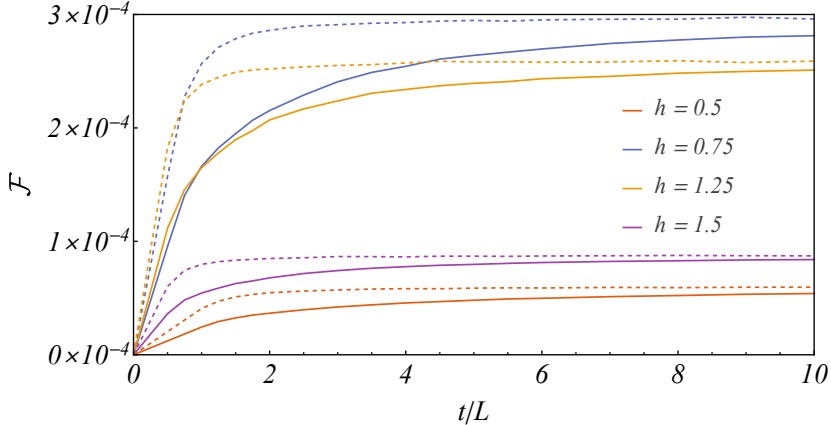

Figure 3: Quasi-particle growth for an interval, for different values of the transverse field $h$ (different colors). Full lines are the multiplets predictions as expected from Eq. (40) whilst dashed lines are the results when pair-structure is considered. As expected, they are both approaching the same saturation value for large time.

quasi-particle picture can be cast in a scaling form similar to Eq. (37)

$$\mathcal{S}_{[0,L]}^{(N)} = \frac{N}{1-N} L\left[\gamma^2 \mathcal{F}(tL^{-1}, h) + \mathcal{O}(\gamma^3)\right]. \tag{40}$$

The function $\mathcal{F}$ can be read directly from Eq. (36), resulting in an expression similar to Eq. (38), albeit more complicated and bearing a $t/L$ dependence.

For times $t \leq L/(2v_M)$ with $v_M = \max_{k \in [-\pi, \pi]} |v(k)|$, $\mathcal{F}(tL^{-1}, h)$ grows linearly $L\mathcal{F}(tL^{-1}, h) = 2tF(h)$ with $F(h)$ the half-line prediction (38). On the contrary, at large times it saturates to a finite value. From consistency requirements with the local relaxation after a quantum quench, such a steady value must be equal to the thermodynamic entropy computed on the final GGE

$$\lim_{t \to \infty} \mathcal{S}_{[0,L]}^{(N)}(t) = \frac{L}{1-N} \int \frac{dk}{2\pi} \log\left[(1-n(k))^N + n^N(k)\right]. \tag{41}$$

Eq. (41) is exact: checking its consistency with Eq. (40) using Eq. (39) is a straightforward calculation.

Accessing the scaling part of the entanglement entropies for finite intervals by mean of iTEBD methods is difficult: in Fig. 3 we provide a plot of $\mathcal{F}(tL^{-1}, h)$ for different values of the magnetic field. Again, on top of the correct multiplet result, we plot what one would get employing the mode density $n(k)$ into the ansatz based on the pair structure. The curves are clearly far apart, confirming once again the importance of considering the multiparticle structure in the quasi-particle ansatz Eqs. (8-9).

It should be stressed that, for the current choice of the $C_i$ parameters, the quasiparticle ansatz based on the pair structure is not justified at all, since only quartuplets are present. For a generic choice of the couplings $C_i$, also paired excitations will be present and the standard ansatz is expected to improve. However, for the sake of dealing with more compact expressions, we consider only the choice Eqs. (24-25).

# 4 Conclusions

In this paper we investigated the applicability of the quasi-particle ansatz for the entanglement growth in those frameworks exiling the commonly-assumed pairwise structure of the excitations' pattern. Indeed, there are several quenches exhibiting a far richer structure than the simplest paired one: this is verified, for example, in any quench from the ground state of an interacting model (on which the Wick theorem does not hold) to a free one, where the paired-excitation pattern implies the Wick theorem. As a specific example, we considered a sudden homogeneous quench from a weakly interacting spin chain to a free one, unveiling the rich structure of the initial state in terms of the post-quench modes, where excitations' multiplets beyond simple pairs are present. We generalize the quasi-particle picture to this wide class of states, capturing the growth (in the scaling limit) of the Rényi entropies of integer index $N$. Albeit the complicated structure of the initial state makes difficult to exactly determine the input of the quasi-particle ansatz, we provided a systematic expansion for small interactions which is in excellent agreement with exact iTEBD simulations.

Our findings confirm the wide applicability of the quasi-particle ansatz to our understanding of entanglement spreading, pointing out at the same time the urge of going beyond the simple pair-structure.

Several interesting developments are left for future investigations. First of all, the quasi-particle picture has been proven to be a precious tool in investigating the correlation functions as well [68]: it would be interesting to explore the consequences of higher multiplets in this analysis.

While we accessed the Rényi entropies for integers $N$, the von Neumann entanglement entropy still remains an elusive quantity, the reason being the impossibility of performing an analytic continuation $N \to 1$ term by term in the small coupling expansion. In this perspective, partial resummations of the perturbative series could lead to meaningful analytic continuations: a better understanding of the pattern of the diagrammatic representation will be surely helpful in this perspective.

Very recently, an ab-initio technique to address entanglement entropies, based on form-factors of twist fields, has been proposed in Ref. [73] and successfully applied to the Ising field theory, starting from an initial state with pair-structure. It would be of great interest applying those techniques to the more general form of states considered in this paper.

Lastly, a surely compelling question concerns quenches towards truly interacting integrable models: as clearly shown by the form-factor perturbation theory of the pre-quench state [74], quenching towards an interacting integrable model will generally result in a much more complicated structure than pairwise excitations. The lack of a pair-structure in generic interaction quenches in integrable models have been pointed out in Ref. [75], with the further development of a perturbative study in terms of the pre-quench dynamics [75–77]. However, the question whether states in the form Eq. (10), together with the entire quasi-particle picture, can be generalized to truly interacting integrable case is still an open problem, which we leave for future investigations.

# 5 Acknowledgements

We thank P. Calabrese and V. Alba for useful comments on the manuscript.

**Funding information**  This work has been supported by the European Research Council under the ERC Advanced grant 743032 DYNAMINT.

# A   The diagrammatic expansion for the pre-quench state

In this appendix we discuss a systematic description of Eq. (22) in terms of Feynman diagrams, whose iterative solution ultimately leads to a power-expansion of the $\mathcal{K}$ wavefunction in terms of the small parameter $\gamma$. This representation was already discussed in Ref. [71] for the bosonic case and can be promptly generalized to the fermionic one. First of all, we represent the wavefunction $\mathcal{K}_{2n}$ with a dot with $2n$ departing arrows, each of them representing a momentum of the wavefunction. We use a similar representation for the interaction vertexes $\mathcal{H}_{n,m}$, assigning them empty circles with $n$ departing and $m$ incoming arrows, representing the momenta in the vertex.

The value of $\mathcal{I}_{2n}$ in Eq. (22) is then computed according with the rules below. As a guide for better understanding the Feynman rules, in Fig. 4 we provide a specific example of a possible diagram and its value.

1. Draw a single vertex associated with $\mathcal{H}_{n',m'}$ and a few vertexes associated with $\mathcal{K}_{2n''}$. Then, contract the incoming $m'$ arrows of the vertex $\mathcal{H}_{n',m'}$ with the arrows departing from the $\mathcal{K}-$vertexes. All the in-going arrows must be contracted, the resulting diagram must not have disconnected parts and there must be exactly $n$ out-going legs.

2. At any vertex one must impose the conservation of momenta, according with the direction of the arrows. A global Dirac$-\delta$ for the conservation of the $n-$outgoing momenta always appears and must be removed (as it should be clear from Eq. (21), where the global $\delta-$function is explicitly factorized).

3. Extra caution must be payed because of the fermionic nature of the operators, giving the correct sign to the diagram. This can be done as follows: write the vertex $\mathcal{H}_{n',m'}$ and, after that, the product of the wavefunctions appearing in the diagram. Then, as per the Wick theorem, add the sign of the permutation needed to reshuffle the position of the momenta in order to contract those that are equal and put the out-going one into the right order.

4. One must integrate over the momenta carried by the incoming legs and divide by a symmetry factor equal to the number of permutations of legs and vertexes that leave the diagram the same.

5. Finally, sum over all the possible diagrams with $n-$outgoing legs.

In general, if the interaction vertexes $\mathcal{H}_{n,n'}$ with $n+n' > 2$ are present, the equations (22) cannot be closed with a finite number of wavefunctions $\mathcal{K}$: an important exception is the case when only vertexes $n+n' = 2$ are present. In this case, we are considering a quench among quadratic, i.e. free, models where the squeezed state form Eq. (6) must be recovered. It is instructive to approach the problem within this formalism, providing in the meanwhile a consistency check: let us assume only the interactions $\mathcal{H}_{2,0}$, $\mathcal{H}_{1,1}$, $\mathcal{H}_{0,2}$ are present. In Fig. 5 we draw the diagrams relevant for the case $n = 1$ of Eq. (22), where we already assumed $\mathcal{K}_{2n\neq 2} = 0$. These graphs lead to the following non-linear equation for $\mathcal{K}_2$

$$\omega(k)\mathcal{K}_2(k,-k) + \frac{1}{2}\gamma\mathcal{H}_{2,0}(k,-k|) + \gamma\mathcal{H}_{1,1}(k|k)\mathcal{K}_2(k,-k)$$
$$-\gamma\frac{1}{2}\mathcal{H}_{0,2}(|-k,k)\mathcal{K}_2(-k,k)\mathcal{K}_2(k,-k) = 0, \tag{42}$$

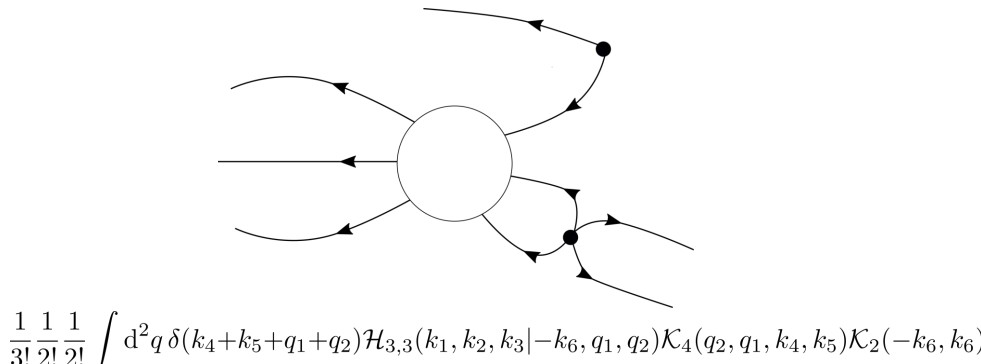

$$\frac{1}{3!}\frac{1}{2!}\frac{1}{2!}\int d^2q\,\delta(k_4+k_5+q_1+q_2)\mathcal{H}_{3,3}(k_1,k_2,k_3|-k_6,q_1,q_2)\mathcal{K}_4(q_2,q_1,k_4,k_5)\mathcal{K}_2(-k_6,k_6)$$

Figure 4: Above, we provide an example of a Feynman diagram contributing to $\mathcal{I}_6(k_1,...,k_6)$, together with its value. The symmetry factor in front of the integral comes from the possible permutations of legs: more specifically, the factorials 3! and 2! come from the permutation of the triplet and pair of external legs, while an additional term 2! is due to the permutations of the internal legs.

which has two possible solutions

$$
\begin{aligned}
\mathcal{K}_2(k,-k) \;=\; & \frac{1}{\gamma\mathcal{H}_{0,2}(|-k,k)}\Bigg[\omega(k)+\gamma\mathcal{H}_{1,1}(k|k) \\
& \pm\sqrt{(\omega(k)+\gamma\mathcal{H}_{1,1}(k|k))^2+\gamma^2|\mathcal{H}_{0,2}(|k,-k)|^2}\;\Bigg].
\end{aligned}
\tag{43}
$$

The correct choice is the minus sign, which ensures that in the $\gamma\to 0$ limit the amplitude $\mathcal{K}_2$ vanishes. Moreover, plugging the two solutions into Eq. (22) and looking at the case $n=0$, one can check that choosing the minus sign in Eq. (43) corresponds to the minimum energy $E_G$. Using in the above equation the amplitudes associated with quenches of the magnetic field in the Ising model (18), one promptly recovers the known result obtained by mean of Bogoliubov rotations [67].

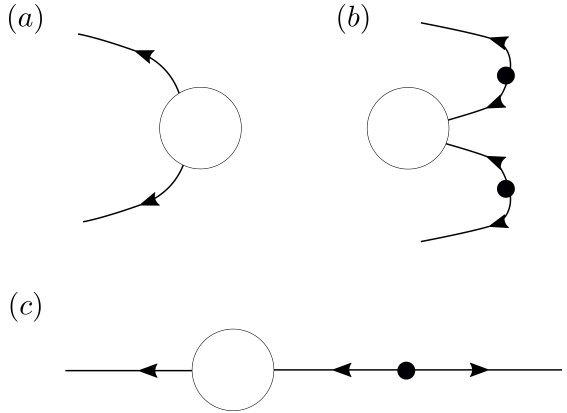

Figure 5: The diagrammatic representation of the terms in Eq. (42).

# B  The diagrammatic expansion for the entanglement spreading

In this appendix we present the systematic expansion of the quasi-particle ansatz which has, as a first non trivial order, the result Eq. (36).

   We will proceed through a direct computation of the reduced density matrix and powers thereof, applying the definition Eq. (35): as we already commented, states in the form (30) seem unfeasible of exact computations, but an expansion valid for small $\mathcal{K}$ can be straightforwardly performed through Feynman diagrams, as we are going to discuss. Rather than considering the state Eq. (30), it is more convenient to use the unnormalized version

$$|\Psi_x\rangle = \exp\left[\sum_{n=1}^{\infty} \frac{1}{(2n)!} \int_{-\pi}^{\pi} \mathrm{d}^{2n}k\, \delta\left(\sum_{j=1}^{2n} k_j\right) \mathcal{K}_{2n}(k_1, ..., k_{2n}) \prod_{j=1}^{2n} \eta_x^{\dagger}(k_j)\right] |0_x\rangle. \qquad (44)$$

In this way, the norm appearing in Eq. (30) is $\mathcal{N} = \langle\Psi_x|\Psi_x\rangle$. As a warm up, we start presenting the Feynman rules for computing the norm of the state, then generalize them to the partial traces and powers of the reduced density matrices. Expanding the exponential in Eq. (44) and contracting with the bra, one contracts the momenta in the wavefunctions $\mathcal{K}_{2n}$ coming from the ket with the conjugated $\mathcal{K}_{2n}^*$ coming from the bra. With the same notation of the previous appendix, we represent the $\mathcal{K}_{2n}$ wavefunctions with vertexes with $2n$ outgoing arrows. On the other hand, the wavefunctions $\mathcal{K}_{2n}^*$ are represented as vertexes with $2n$ ingoing arrows: performing the contractions, only legs with compatible orientation can be joined. While computing the norm $\mathcal{N}$, all the legs must be contracted: one should then integrate over the momenta associated with internal legs enforcing momentum conservation at each vertex. Then, one must divide for a symmetry factor equal to the number of permutations of legs and vertexes that leave the diagram unchanged. Lastly, the sum over all the possible diagrams must be considered: in this respect, as it is standard in perturbative expansions of partition functions, the sum over the Feynman diagrams can be represented as an exponential of the sum over the connected diagrams

$$\mathcal{N} = \sum\left[\text{Feynman diagrams}\right] = \exp\left[\sum\left[\text{connected Feynman diagrams}\right]\right]. \qquad (45)$$

In writing this expression, a subtlety arises. Indeed, in any connected Feynman diagram the global momentum conservation is always satisfied, resulting in a formally divergent contribution $\delta(0)$, where the Dirac-$\delta$ acts in the momentum space. As it is standard in these computations, one can make sense of this divergence by mean of a finite-volume regularization (34) of the system, which would lead to the replacement $\delta(0) \rightarrow \ell/(2\pi)$, with $\ell$ the system's size. In this perspective, the logarithm of the norm behaves extensively in the system's size $\log\langle\Psi_x|\Psi_x\rangle \propto \ell$: this is in agreement with the fact that $\langle\Psi_x|\Psi_x\rangle$ can be interpreted as a partition function [71] and hence its logarithm, namely a free energy, is extensive.

   We can now generalize the method to compute reduced density matrices, then powers thereof. Again, we work with the unnormalized state Eq. (44) and define the unnormalized reduced density matrix $\hat{\sigma}_{\mathcal{A}x,t}$ as

$$\hat{\sigma}_{\mathcal{A}_{x,t}} = \text{Tr}_{\mathcal{B}_{x,t}} |\Psi_x\rangle\langle\Psi_x|. \qquad (46)$$

Note that we clearly have $\text{Tr}_{\mathcal{A}_{x,t}} \hat{\sigma}_{\mathcal{A}_{x,t}} = \mathcal{N}$, therefore the normalized density matrix is $\hat{\rho}_{\mathcal{A}_{x,t}} = \hat{\sigma}_{\mathcal{A}_{x,t}}/\mathcal{N}$ and the Rényi entropies are

$$s_{\mathcal{A}_{x,t}}^{(N)} = \ell^{-1}\left(\frac{1}{1-N} \log \text{Tr}_{\mathcal{A}_{x,t}} \hat{\sigma}_{\mathcal{A}_{x,t}}^N - \frac{N}{1-N} \log\mathcal{N}\right). \qquad (47)$$

The Feynman rules to compute Eq. (46) are exactly the same of those for the norm $\mathcal{N}$, with two exceptions: *i)* we can now have free (namely, uncontracted) external legs representing the

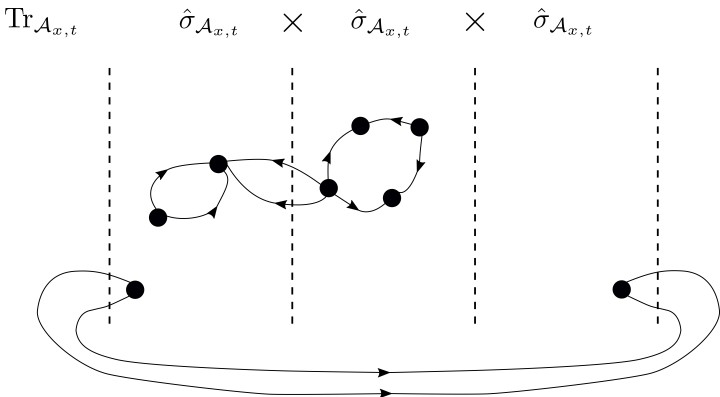

Figure 6: An example of the possible Feynman diagrams for $\mathrm{Tr}_{\mathcal{A}_{x,t}} \hat{\sigma}^3_{\mathcal{A}_{x,t}}$.

degrees of freedom of $\mathcal{A}_{x,t}$ (on which we are not tracing yet) and *ii)* the momenta associated with the contracted legs on which we are integrating are confined to the domain $\mathcal{B}_{x,t}$, rather than to the whole Brillouin zone. Again, the sum over the Feynman diagrams can be exponentiated in terms of the connected diagrams, as in Eq. (45): this exponential of diagrams represent the reduced density matrix Eq. (46).

Armed with this machinery, we can now discuss powers of the reduced density matrix. Referring to Fig. 6, we divide the plane into different sectors by mean of vertical dashed lines, each sector represents a reduced density matrix. In particular, while computing $\hat{\sigma}^N_{\mathcal{A}_{x,t}}$, one needs $N+1$ vertical lines, splitting the plane in $N$ parts). The diagrams drawn in each region belong to the respective density matrix in the product, which is then performed contracting the free legs with the following rules. Legs can be connected only between adiacent regions and outgoing arrows can be contracted only towards the left, while ingoing arrows only towards the right: this because in(out) arrows are respectively associated with bra(ket) states. The internal legs crossing the vertical lines represent contraction among different density matrices: therefore, their momenta are integrated over the $\mathcal{A}_{x,t}$ domain. After this operation, all the legs are contracted, with the exceptions of the out-going legs in the leftmost region and in-going arrows in the rightmost one. Now, we can finally take the trace of the product joining together the legs in the leftmost and rightmost regions, their momenta being still integrated over the $\mathcal{A}_{x,t}$ domain. As usual, momentum conservation must be imposed at any vertex and the proper symmetry factors kept into account.

Finally, the sum over all the possible diagrams must be considered. Again, the sum over the Feynman diagrams is equivalent to the exponential of the sum over the connected diagrams.

$$\mathrm{Tr}_{\mathcal{A}_{x,t}} \hat{\sigma}^N_{\mathcal{A}_{x,t}} = \exp\left[\sum\left[\text{connected Feynman diagrams}\right]\right], \tag{48}$$

and, as we have already discussed while considering the norm $\mathcal{N}$, a global momentum conservation results in a divergent term $\delta(0)$, which is regularized with a finite volume scheme $\delta(0) \to \ell/(2\pi)$. Hence, $\log \mathrm{Tr}_{\mathcal{A}_{x,t}} \hat{\sigma}^N_{\mathcal{A}_{x,t}} \propto \ell$ and the density of entanglement $s^{(N)}_{\mathcal{A}_{x,t}}$ in Eq. (47) is well defined and independent from $\ell$. Now, we present the computation at first order in the small parameter $\gamma$: in view of Eq. (23), we keep up to $\mathcal{O}(\mathcal{K}^2)$. At this order, the logarithm of the norm is simply

$$\log\mathcal{N} = \delta(0)\sum_n \frac{1}{(2n)!}\int_{-\pi}^{\pi} \mathrm{d}^{2n}k\,\delta\left(\sum_{i=1}^{2n} k_i\right)|\mathcal{K}_{2n}(k_1,...,k_{2n})|^2 + \dots \tag{49}$$

Above, the divergent prefactor is regularized as we have already discussed $\delta(0) \to \ell/(2\pi)$.

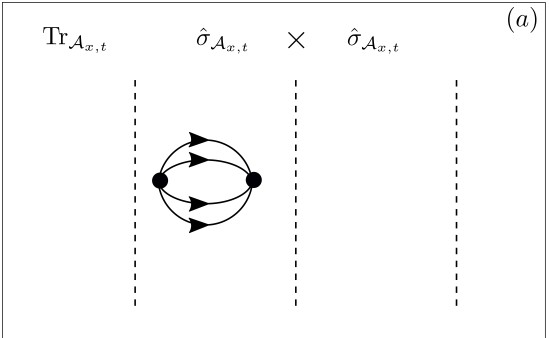
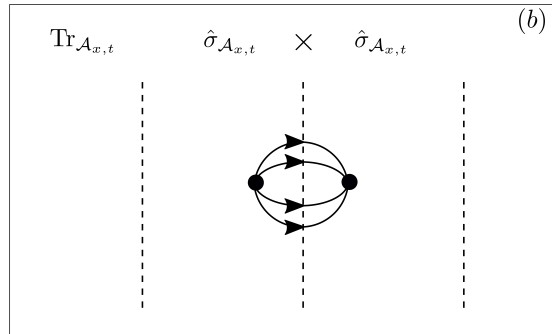

Figure 7: The connected Feynman diagrams contributing to $\mathrm{Tr}_{\mathcal{A}_{x,t}} \hat{\sigma}^N_{\mathcal{A}_{x,t}}$ up to $\mathcal{O}(\mathcal{K}^2)$, here depicted for $N = 2$. Panels $(a)$ and $(b)$ respectively represent the first and second line in Eq. (50): indeed, the internal arrows in Panel $(a)$ are integrated over the $\mathcal{B}_{x,t}$ domain, while those in Panel $(b)$ on the domain $\mathcal{A}_{x,t}$. The symmetry in the replica space allows to move the diagram across the replicas, resulting in an overall $N$ factor. For example, in Panel $(a)$ the diagram is drawn in the first region, but it can be put in the second one as well leading to the same contribution, thus accounting for a global factor 2.

For what concerns $\mathrm{Tr}_{\mathcal{A}_{x,t}} \hat{\sigma}^N_{\mathcal{A}_{x,t}}$ with $N \geq 2$ one gets (see Fig. 7)

$$\log \mathrm{Tr}_{\mathcal{A}_{x,t}} \hat{\sigma}^N_{\mathcal{A}_{x,t}} = \delta(0) \frac{N}{(2n)!} \int_{\mathcal{B}_{x,t}} \mathrm{d}^{2n}k\, \delta\left(\sum_{i=1}^{2n} k_i\right) |\mathcal{K}_{2n}(k_1, ..., k_{2n})|^2$$

$$+ \delta(0) \frac{N}{(2n)!} \int_{\mathcal{A}_{x,t}} \mathrm{d}^{2n}k\, \delta\left(\sum_{i=1}^{2n} k_i\right) |\mathcal{K}_{2n}(k_1, ..., k_{2n})|^2 + ... \quad (50)$$

Above, the $N$ prefactor is due to the symmetry in the replica space (see again Fig. 7). Plugging this result in Eq. (47), one immediately recovers Eq. (36).

## C  Numerical simulation of time evolution

The numerical simulations of the entanglement dynamics has been obtained by using the infinite volume Time-Evolving Block-Decimation (iTEBD) algorithm [78]. The algorithm represents a generic translational invariant state in one dimension as

$$|\Psi\rangle = \sum_{..., s_j, s_{j+1}, ...} \cdots \Lambda_o \Gamma_o^{s_j} \Lambda_e \Gamma_e^{s_{j+1}} \cdots |..., s_j, s_{j+1}, ...\rangle, \quad (51)$$

where $s_j$ spans the local spin-1/2 Hilbert space, $\Gamma_{o/e}^s$ are $\chi \times \chi$ matrices associated with the odd/even lattice site; $\Lambda_{o/e}$ are diagonal $\chi \times \chi$ matrices containing the singular values corresponding to the bipartition of the system at the odd/even bond.

Exploiting the representation (51), the Rényi entropies of the semi-infinite subsystem are given by

$$S^{(N)}_{[0,\infty]} = \frac{1}{1-N} \log \mathrm{Tr}[(\Lambda^\dagger_{o/e} \Lambda_{o/e})^N], \quad (52)$$

where a bipartition at the odd/even bond can be indifferently chosen.

The infinite Matrix Product State (iMPS) representation of the ground state $|\Psi_{GS}\rangle$ of the pre-quench $XYZ$ Hamiltonian has been obtained by (imaginary)time-evolving the following

initial product state

$$
|\Psi_0\rangle =
\begin{cases}
\otimes |\uparrow_x\rangle & h < 1 \\
\otimes |\uparrow_z\rangle & h > 1
\end{cases}.
\tag{53}
$$

We used a second-order Suzuki-Trotter decomposition of the evolution operator with imaginary time Trotter step $\tau = 10^{-5}$. The pre-quench parameters $\{C_1, C_2, C_3, C_4\}$ have been tuned accordingly to Eqs. (24) (25), with $\gamma \in \{0.1, 0.08, 0.06\}$ and transverse field $h \in \{0.1, 0.2, 0.3, \ldots, 1.9\}$. Thanks to the energy gap between the ground state and the the rest of the spectrum, the iMPS ansatz is very accurate (up to machine precision) by retaining an auxiliary dimension $\chi_0 = 16$ or $32$, depending on the vicinity of the critical point (i.e. $\gamma = 0$, $h = 1$).

Similarly, the post-quench time evolution has been obtained by evolving the corresponding $|\Psi_{GS}\rangle$ with the Ising Hamiltonian with $\gamma = 0$. For this purpose, a second-order Suzuki-Trotter decomposition of the evolution operator was used again, with real time Trotter step $\mathrm{d}t = 10^{-3}$. In order to keep the truncation error as small as possible ($\leq 10^{-6}$), the auxiliary dimension was allowed to grow up to $\chi_{MAX} = 1024$ which was sufficient to reach a maximum time $T = 100$. We were able to reach such large times thanks to the relatively small quenches considered. As explained in the main text, the entanglement entropy, for all quenches under investigation, growths linearly with a slope proportional to $\gamma^2$, thus remaining very small for all the duration of the simulations.

The numerical data for the Rényi entropies as a function of the rescaled time $\gamma^2 t$ showed a linear increase (apart from oscillations) whose slope depends on the particular value of the transverse field $h$. In particular, a numerical estimation of the entanglement entropy slope $\partial_{\gamma^2 t} S^{(N)}_{[0,\infty]}$ has been obtained by performing a linear fit of the iTEBD data in the entire time-window $0 \leq t \leq 100$, for the three different values of $\gamma$ we have considered. Thereafter, the $\gamma \to 0$ value has been evaluated by linear extrapolation.

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
