# Peer review of "Entanglement spreading and quasiparticle picture beyond the pair structure"

_SciPost Physics, doi:SciPost Phys. 8, 045 (2020)_

## Round 2 · Referee Report · Anonymous (Referee 1) · 2020-2-29

Strengths

1- Provides some new analytical results concerning a difficult and interesting problem (entanglement entropy growth after a quantum quench in a 1d interacting problem). 2- The above results go beyond the simple "pairwise-created quasiparticle" picture, and are checked with iTEBD numerics (on an Ising spin chain).

Weaknesses

1- The numerical part is quite short and could have been developed further.

Report

This paper presents a theoretical study of the growth of quantum entanglement after a quantum quench. The specific situation that is treated is that of a weakly-interacting one-dimensional system that is quenched to a free-fermion Hamiltonian. It is generally useful to describe the initial state in terms of the elementary excitations of the post-quench Hamiltonian. In the simplest situations (such as free-to-free quenches) the initial state can be described in terms of excitations created (only) in pairs. In such situations where the pairs are uncorrelated, the use of the Wick theorem allows to compute the entanglement entropies. However, in more general situations, higher "multiplets" of excitations are also created (as described in Ref. [71]), and the calculation of the entropy becomes very nontrivial. This manuscript deals with this more general case, by focusing on the limit where the pre-quench Hamiltonian is weakly interacting. In that limit the authors obtain perturbatively an expression for the Rényi entanglement entropy of a subsystem, as a function of time and of the initial state decomposition in terms of quasiparticles (Eq. 36). The authors then consider a specific quench in a 1d quantum Ising chain and compute the associated entropy rate (Eq. 38). This results is then checked with direct numerical simulations, using the iTEBD algorithm.

This paper is well writte and contains some new and interesting results. I would recommend it for publication in SciPost after the authors have addressed the questions below (as well as the "Requested changes").

  • The entropies shown in Fig. 2 (as well as in Fig. 3) appear to be extremely small (at most of the order or 10^(-5) or 10^(-4)). Is the present perturbative approach limited to such tiny entropy values ? If not, the authors could try to illustrate their approach with examples where the entropy growth is quantitatively more larger.

  • The right panel of Fig. 2 shows that the calculations using the generalized quasi-particle picture are correct, but it also indicates that the simpler pair-ansatz is a relatively good approximation. Given the fact that the parameters of the quench were precisely chosen to suppress the production of pairs of quasiparticles, we might expect that in generic situations the pair-ansatz would be an even better approximation. Can the author comment/elaborate on this ?

Requested changes

  • The Eq. (5) does not seem to be an extensive energy. Missing system-size factor ?
  • A few details about the derivation of Eq. (29) would be useful.
  • The Section 3 explains how to compute entanglement entropies using the quasi-particle picture. The title of this section could therefore be a bit more explicit, for instance by mentioning "entanglement entropy".
  • Going from Eq. 36 to eq. 38 certainly requires a few intermediate steps. Giving a few additional calculation details would be useful.
  • Due to the tiny linewidths in the right panels of Fig. 2 the colored lines are difficult to distinguish. It is also not very aesthetic to have large fonts in the right panels, and small ones in the left one.
  • No details are given about the numerics. It would be useful to provide the readers with some informations about the iTEBD implementation and the simulations parameters (bond dimension and/or discarded weight, time step, possible convergence checks). These precision issues seem all the more important as the data displayed in Fig. 2 are obtained by substracting O(1) entropies (at time t and time 0) which are very close to each other.

---

## Round 2 · Referee Report · Anonymous (Referee 2) · 2020-3-3

Strengths

  • quench results for genuinely interacting initial Hamiltonians
  • very good agreement of perturbation theory with numerics

Weaknesses

  • quench example considered is somewhat artificial

Report

The authors study the growth of entanglement after a quench
where the initial state does not allow for a description
in terms of simple quasiparticle-pair excitations.
In particular, they consider the case with a special multiplet
structure, constructed in a translational invariant fashion,
such that more than two entangled particles are emitted
at every spatial location. A generalized quasi-particle
picture is then developed and applied for the calculation
of the Renyi entropy in a perturbative manner, i.e. in a
power series expansion of the multiplet creation amplitudes.
Unfortunately, the von Neumann entropy can not be obtained
in this way, since the perturbation expansion does not
commute with the analytic continuation of the Renyi index.

The leading order result for the Renyi entropy is obtained
explicitly and tested on a specific lattice model, namely a
weakly perturbed Ising chain. The quench is performed from
the ground state of the weakly interacting chain towards the
free-fermion point. The authors carry out numerical iTEBD
simulations which nicely confirm the first order result,
shown to be clearly distinct from the pair-structure ansatz.

The manuscript deals with an interesting problem and the
results are novel and sound, I believe that they deserve
publication in Scipost Physics. I have only some minor
issues which the authors should address.

Requested changes

I. In order to test the prediction of the generalized quasiparticle picture, the authors use a lattice Hamiltonian that is heavily fine-tuned in order to suppress the pair-creation amplitude. I believe this is chosen such that the pair/multiplet ansatz curves for the entropy production be better distinguishable. However, I'm missing some comment about the generic situation, i.e. when there are simultaneous pair and quadruplet production. Is the effect of the quadruplets very small in this case? Or otherwise, can the authors think of a realistic quench scenario where the pair-production can be suppressed without fine-tuning of the couplings?

II. The manuscript is full of typos and sentences with incorrect grammar. Below only a few examples I noticed. However, I would strongly recommend a thorough proofreading of the full text.

  1. ansätz --> ansatz (throughout entire manuscript)
  2. Von Neumann --> von Neumann (throughout entire manuscript)
  3. "being it either classical or quantum" ?? (in introduction)
  4. "provides a net and clear framework" ?? (in introduction)
  5. "obtained tracing out" --> obtained by tracing out (after Eq. (1))
  6. "this is in not true in" (after Eq. (9))
  7. Axis labels of Fig. 2 left: the (N) should be in superscript

---

## Round 3 · Referee Report · Anonymous (Referee 1) · 2020-3-17

Report

The authors have answered all the points raised in my first report, and they have fixed a number of issues. I think that the paper is now suitable for publication in SciPost.

---

## Round 3 · Referee Report · Anonymous (Referee 2) · 2020-3-17

Report

The authors have properly addressed all the issues in the referee report and made changes to the manuscript accordingly, the revised version can be accepted for publication.

---

## Round 3 · Author Response

We thank the referees for their appreciation of our work and the positive comment we received. A few minor suggestions have been pointed out to further improve our manuscript.
Hereafter, we separately address the questions asked by the two referees and enlist the changes we made.

Referee 1:

Q. The entropies shown in Fig. 2 (as well as in Fig. 3) appear to be extremely small (at most of the order or 10^(-5) or 10^(-4)). Is the present perturbative approach limited to such tiny entropy values? If not, the authors could try to illustrate their approach with examples where the entropy growth is quantitatively larger.

A. First of all, let us point out that the quasiparticle picture captures the scaling part of the entanglement: going to larger systems and longer times, the entanglement can be made, in principle, arbitrary large. Therefore, one should rather focus on the rate of entanglement growth. Our approach provides an expansion for this rate in the case of small interaction quenches, which is therefore perturbative in the interaction strength as it is clear, for example, from Eq. (37).
Still referring to Eq. (37), the F(h) function is the growth rate (at the leading order in perturbation theory) renormalized on the coupling constant: in our model F(h) turns out to be small (Fig. 2), as the referee pointed out. We believe this is a feature of the model we considered and the situation could be different in other systems, however the excellent agreement between our prediction and the numerical data already confirms the validity of our approach, therefore we do not think that analyzing other models will qualitatively improve our work.

Q. The right panel of Fig. 2 shows that the calculations using the generalized quasi-particle picture are correct, but it also indicates that the simpler pair-ansatz is a relatively good approximation. Given the fact that the parameters of the quench were precisely chosen to suppress the production of pairs of quasiparticles, we might expect that in generic situations the pair-ansatz would be an even better approximation. Can the author comment/elaborate on this?

A. In the revised version of the manuscript here submitted, we explicitly comment on this point above Eq. (37) and at the end of Section III. First, let us point out that the pair ansatz, with the choice of couplings C_i we made in our work, is not in principle justified, since based on the wrong assumption that excitations are excited in pairs (while the pairs are absent at first order in the perturbative expansion). However, one could naively measure the final momentum distribution and, blindly assuming the state has pairwise excitations, feed the momentum distribution to the pair-ansatz and try to compare with the numerics: this is the scenario we wanted to describe. However, if one does so tangible discrepancies are found, as shown in Fig. 2: for values of the magnetic field close to the criticality, between the correct result and the wrong ansatz there is almost a factor 2.
Now, let us consider the generic case, which is now shortly commented in the text: for generic values of the coupling C_i, both pairs and quartuplets will be excited at the same order in the perturbation theory and, as Eq. (36) clearly states, they will separately contribute to the entanglement. Therefore, a quasiparticle ansatz based on the pair structure should at least capture part of the entanglement growth, probably resulting in a better approximation of the actual entanglement growth.

Requested changes:

Q. The Eq. (5) does not seem to be an extensive energy. Missing system-size factor ?
A. There are not system-size factor missing, because of the normalization of the mode operators. Indeed, they are correlated as <\eta^\dagger(k)\eta(q)>=delta(k-q) n(k) with n(k) the mode density (this equation appears in-line right before of Eq. (7)). In the energy, there are terms \eta^\dagger(k)\eta(k), with the consequence that, after the expectation value is taken, a \delta(0) term appears: this is interpreted as the system size, once a proper finite-size regularization is adopted, as it is customary in field theories calculations. Indeed, the same extensive prefactor appears also in the computations of the Renyi entropies, as it is explained in Eq. (34).

Q. A few details about the derivation of Eq. (29) would be useful.
A. Right above Eq. (24), we now explain in words which are the steps to reach Eq. (29). These are lengthy, but trivial, algebraic manipulations that we prefer to not report explicitly for the sake of a lighter exposition.

Q. The Section 3 explains how to compute entanglement entropies using the quasi-particle picture. The title of this section could therefore be a bit more explicit, for instance by mentioning "entanglement entropy".
A. We changed the title accordingly.

Q. Going from Eq. 36 to eq. 38 certainly requires a few intermediate steps. Giving a few additional calculation details would be useful.
A. Right before of Eq. (37), we explain in words how to pass from Eq. (36) to Eq. (38).

Q. Due to the tiny linewidths in the right panels of Fig. 2 the colored lines are difficult to distinguish. It is also not very aesthetic to have large fonts in the right panels, and small ones in the left one.
A. We increased the thickness of the lines to help to distinguish the colors and we corrected the font as requested.

Q. No details are given about the numerics. It would be useful to provide the readers with some information about the iTEBD implementation and the simulations parameters (bond dimension and/or discarded weight, time step, possible convergence checks). These precision issues seem all the more important as the data displayed in Fig. 2 are obtained by substracting O(1) entropies (at time t and time 0) which are very close to each other.
A. We added a short appendix with the details of the iTEBD simulations

Referee 2:

Q. In order to test the prediction of the generalized quasiparticle picture, the authors use a lattice Hamiltonian that is heavily fine-tuned in order to suppress the pair-creation amplitude. I believe this is chosen such that the pair/multiplet ansatz curves for the entropy production be better distinguishable.
However, I'm missing some comments about the generic situation, i.e. when there are simultaneous pair and quadruplet production.
Is the effect of the quadruplets very small in this case? Or otherwise, can the authors think of a realistic quench scenario where the pair-production can be suppressed without fine-tuning of the couplings?
A. This question is very similar to one of the requests of Referee 1, therefore we modified the manuscript taking into account both the suggestions. In the improved version of our manuscript, we comment on the general situation where both pairs and quadruplets are present, which is the generic situation where the C_i coefficients are not fine-tuned. In this case, both pairs and quadruplets are created at the same order in the perturbation theory and they will separately contribute to the entanglement, as it is clear in Eq. (36). In general, an erroneous quasiparticle ansatz based only on pairwise excitation, will probably better capture the general case where pairs are present, but the corrections due to quadruplets will be always there. In principle, the generic result can be read from Eq. (36) and Eq. (23), while the H_{n,n'} coefficients can be obtained as explained in Sec 2.1 and App. A. However, the general case would result in quite lengthy expressions without any qualitative change in our findings, therefore we rather prefer to focus on a quench which clearly needs a multiplet generalization of the quasiparticle ansatz.

Q. The manuscript is full of typos and sentences with incorrect grammar. Below only a few examples I noticed. However, I would strongly recommend a thorough proofreading of the full text.
A. The manuscript has been carefully edited and the typos fixed.

---

## Round 3 · List of Changes

• A short discussion on the iTEBD algorithm has been added as an appendix
  • Extra comments on the case where pairs and quadruplets are both present have been added
  • Extra comments on some mathematical derivations are provided
  • Fig. 2 has been improved
  • Typos fixed

---

## Editorial Decision

published